# Material Analysis of the Remains of a Wooden Chest from the 4th Century and a Proposal for Its Reconstruction

**DOI:** 10.3390/ma16010133

**Published:** 2022-12-23

**Authors:** Rebeka Rudolf, Janez Slapnik, Rajko Bobovnik

**Affiliations:** 1Faculty of Mechanical Engineering, University of Maribor, Smetanova ulica 17, 2000 Maribor, Slovenia; 2Faculty of Polymer Technology, Ozare 19, 2380 Slovenj Gradec, Slovenia

**Keywords:** chest, brass tile, analysis, characterisation, reconstruction

## Abstract

A stone chest found in 1971 near one of the largest early Christian basilicas in Northern Dalmatia (Croatia) contained brass tiles decorated with various biblical scenes. An archaeological study confirmed the thesis that the fragments of brass tiles are most likely the remains of a wooden chest made in the 4th century AD, and that this is one of the best preserved archaeological finds of its kind in the world as one of the biblical scenes shows Mary, together with a record of her name (Maria). Based on the preserved brass tiles, a reconstruction of the wooden chest was made in 1973 with tiles glued onto a plastic frame. Subsequent studies have shown that such a reconstruction was not adequate, as some of the brass tiles were destroyed (disintegrated), and they were not connected properly into a whole that could represent the original. For the new reconstruction of this archaeological object it was necessary to carry out a material analysis, including the chemical composition of the brass tiles, as well as to find a solvent for the glue which could be used to remove the brass tiles from the plastic framework without any additional destruction. Based on extensive investigations and material analyses including the following techniques (SEM, EDX, FTIR, DSC), the starting points for the restoration process of the wooden chest with brass tiles were set, as well as the proposal for the appearance of the new chest.

## 1. Introduction

On November 3, 1971, while digging in his garden in the centre of Novalja, Croatia, Vladimir Vidas found a stone chest containing the relics of unknown saints [1]. The site of the find was near one of the largest early Christian basilicas in Northern Dalmatia, the floor of which was adorned with sumptuous mosaics and which can be seen in part even today.

A silver relic decorated with a scene of Christ and the apostles, a silver undecorated relic, a glass vessel and fragments of brass tiles were found in the stone chest. These brass tiles were decorated with various biblical scenes. After an expert examination, it was confirmed that the pieces of brass tiles were, in all probability, the remains of a wooden chest made in the 4th century CE [2]. The analysis showed that this was one of the best preserved archaeological finds of its kind in the world, as one of the biblical scenes depicted Mary, along with a record of her name (Maria). Based on the preserved brass tiles, a reconstruction of the wooden chest (Figure 1a) was made in 1973 by gluing the tiles to a plastic frame. Subsequent archaeological studies in the 1990s have shown that such a reconstruction was not appropriate, as some brass tiles were destroyed (disintegrated), as well as that they were not connected properly into a whole that would represent the original.

A suitable method for removing brass pieces from the plastic box had to be found for a successful reconstruction of the archaeological finds. The brass pieces were glued to the plastic box with an unknown adhesive. First, the composition of this adhesive needed to be determined. Then, a suitable method for reconstruction had to be identified based on the composition of the adhesive. Removal of the brass pieces must not damage or alter the brass in any way. Usually chromatographic or spectroscopic techniques are used for the identification of unknown organic substances. The first use of the spectroscopic analysis in archaeology dates back to 1960, when researchers started to use InfraRed (IR) to address archaeological questions [3]. At the same time, it should be noted that the X-ray Fluorescence (XRF) method of analysis had become well-established, both in the laboratory and industry. The fact that the method is essentially nondestructive makes it particularly attractive for the analysis of archaeological and museum artifacts [4,5]. In the second part, it was necessary to identify the chemical composition of the original brass tiles of the chest. For this purpose, the use of high-resolution Scanning Electron Microscopy (SEM) was foreseen, combined with an analyser for micro-chemical analyses (EDS). SEM has two basic functions: imaging and providing compositional information. As a result, it has been used in archaeological research for almost all archaeological applications in which it is desired to obtain a magnified image of a sample, and/or to determine its composition on a microscopic scale; everything from determining the sources of stone tool raw materials to studying objects more than five millennia old, such as, e.g., old skin of Ötzi the Iceman [6]. In this way it was possible to perform the qualitative and quantitative micro-chemical analyses in a point and on a surface, as well as a qualitative line analysis and surface distribution of elements without destroying the test piece of brass tile. Using this analysis method, we investigated a piece of the original brass tile and adhesive, so that under no circumstances would any damage occur to the original parts of the chest. A demonstration of sampling can be seen in Figure 1b. In the final part, we determined the thermal properties of the adhesive with the help of Differential Dynamic Calorimetry (DSC) [7], and the chemical composition with the help of infrared spectroscopy with Fourier transformation (FTIR) [8]. DSC can detect endothermic and exothermic reactions and provide information about the phase transitions of the material (e.g., melting). Since many substances exhibit unique DSC thermos-grams, it can also be used for the identification of the material [9]. Based on the thermal properties and chemical composition of the adhesive, it was possible to identify suitable solvents that would not react with the brass tiles. Both FTIR spectroscopy and DSC have been used recently to help identify unknown organic compounds in archaeological findings.

## 2. Materials and Methods

### 2.1. Materials

The following materials were used: acetone (Kefo d.o.o, Ljubljana, Slovenia), n-hexane (CARLO ERBA Reagents S.A.S, Val de Reuil Cedex, France), and dichloromethane (CARLO ERBA Reagents S.A.S, Val de Reuil Cedex, France).

### 2.2. Sampling

Taking samples of the brass tiles and glue was carried out under the strict supervision of the staff from the Museum in Zadar (Croatia) in accordance with the Regulation (EC) No. 116/2009. The necessary documents were issued by the Ministry of Culture of the Republic of Croatia, the Administration for the Protection of Cultural Heritage (No. UP/I-612-08/19-02/0034 and UR-BROJ: 532-04-02-13-5-19-4), which made possible the removal of culture goods in EU member states for research.

Using a disinfected blade, four (4) samples of those brass tiles that did not represent a connected sculpture and had a size below 5 mm in at least one dimension were taken manually. Figure 2a shows representative samples of the brass tiles. The glue was removed in several places so that the brass tiles were not damaged. Figure 2b shows such a sample of the removed glue.

### 2.3. Characterisation

#### 2.3.1. XRF

The wt.% of all elements in the brass tiles was measured by XRF (Thermo Scientific Niton XL3t GOLDD+). A Niton XL3t GOLDD + XRF Analyzer equipped with an Ag anode of 50 kV and a 200 μA tube type as a source of X-rays and a Geometrically Optimised Large Area Drift Detector were used for the measurement. The device measures in a wide range of wavelengths of X-rays and for precise chemical composition measurements. A total of 12 XRF measurements were performed on three selected brass tiles as bright, dark and smaller samples.

#### 2.3.2. ICP-MS

The chemical composition of one very small brass tile was measured by Inductively Coupled Plasma-Mass Spectrometry (ICP-MS). The spectrometer used was an HP, Agilent 7500 CE, equipped with a collision cell (Santa Clara, CA, USA). The following conditions were used for the ICP-MS: The power was 1.5 kW, Nebuliser-Meinhard, plasma gas flow was 15 L/min, nebuliser gas flow was 0.85 L/min, make up gas flow was 0.28 L/min and reaction gas flow was 4.0 mL/min. The relative measurement uncertainty was estimated as ±3%. The ICP-MS measurement was performed on only one selected brass tile.

#### 2.3.3. SEM with EDX Analysis

A Scanning Electron Microscope (SEM) Sirion 400NC (FEI, Hillsboro, OR, USA) with an Energy-Dispersive X-ray (EDX) spectroscope INCA 350 (Oxford Instruments, Abingdon, Oxfordshire, UK) was used for the microstructure observation and micro-chemical analyses of the brass tiles. Two methods of investigation were carried out: the first, direct SEM/EDX observation without surface preparation of the dark brass tile; and secondly on the polished surface, which was treated manually with diamond paste 1 µm and felt so that oxide products and other contamination were removed. The requirement was to estimate the chemical composition inside (by volume) the brass tile.

#### 2.3.4. Fourier Transform InfraRed Spectroscopy (FTIR)

The FTIR spectra of the glue samples were recorded using a Spectrum 65 (Perkin Elmer, Waltham, MA, USA) infrared spectrometer using the Attenuated Total Reflectance (ATR) technique. The glue samples were measured from 4000 cm^−1^ to 600 cm^−1^ with a 4 cm^−1^ resolution. The spectra were averaged over 32 scans.

#### 2.3.5. Differential Scanning Calorimetry (DSC)

The thermal properties of the glue samples were determined using a DSC 2 (Mettler Toledo, Greifensee, Switzerland) calorimeter. The samples were measured in a nitrogen (N_2_) atmosphere (20 mL/min) from −70 °C to 200 °C, with a 10 K/min heating/cooling rate. Cooling runs were evaluated first, followed by heating.

### 2.4. Solubility Test

The solubility of the glue samples was tested in three different common organic solvents (acetone, n-hexane, and dichloromethane) by dissolving 0.03 g of the adhesive in 3 mL of solvent at room temperature (23 °C ± 1 °C) for a duration of 5 min.

## 3. Results and Discussions

### 3.1. XRF and ICP-MS Results

#### 3.1.1. Bright Sample—XRF

The results of the XRF chemical analyses presented in Table 1 show that the Pb content in the bright sample of the brass plate was very high and reached almost 13 wt.%, while the Zn content was much lower (6.6 wt.%) compared to the other samples. Ti residues and a rather high Fe content—around 1.2 wt.%—were also identified in this sample. The presence of Ti and other elements can be partly attributed to the casting process, which during the 4th century CE was not yet at such a sophisticated level as today; the starting components were also definitely not sufficiently pure.

#### 3.1.2. Dark and Smaller Samples—XRF

In the case of the darker and smaller samples the chemical composition was comparatively very similar. This was brass with a Zn content slightly above 16 wt.% and Sn between 0.2 and 0.3 wt.%, respectively; Fe was around 0.2 wt.% and Pb between 1 and 2.3 wt.%.

#### 3.1.3. Smaller sample—ICP-MS

The results of the ICP-MS analysis for the smaller sample are shown in the last row of Table 1, and have comparable Sn, Fe and Pb contents to the XRF analysis. A significant variation in chemical composition (almost 50%) was identified for Zn, which can be attributed partly to the ICP-MS method which has a relative variation of ±3%, as well as a possible error achieved by the XRF analysis.

### 3.2. SEM Microstructure with EDX Analysis

#### 3.2.1. SEM/EDX Microscopic Examination of the Unpolished Dark Sample

SEM/EDX analysis was performed to identify the chemical composition further and get a better understanding of chemical properties, as well as the brass surface microstructure. First, a microscopic examination of the surface of the dark sample was performed directly without any preparation. The observation showed that the surface was not smooth, and that there were various impurities and defects on the surface (C-contamination, Al_2_O_3_ oxides, SiC abrasions, holes, etc.). The cause of the presence of the mentioned substances can be attributed to the oxidation, handling and ageing processes. Figure 3 shows the microstructure of this surface with the locations where the three EDX analyses were performed. A total of 3 × 9 = 27 EDX microchemical analyses were performed at the selected sites so that the results could be processed statistically, as shown in Table 2.

The results of the EDX chemical analysis confirmed that there was a high C content on the surface of the sample, which can be attributed to the contamination process. Furthermore, Al and Si were identified, which are typical representatives of oxides and carbides, respectively, which are used as abrasive particles in the process of cleaning various metal surfaces. This suggests that this sample, and, thus, probably all the brass plates covering the wooden chest after its discovery in 1971, were cleaned by a sandblasting process, which presumably damaged these plates partially. The Ca residues may have originated from the soil that was in contact with the chest when it was found. No Cu and Zn were identified on the surface by EDX analysis, indicating that the surface was covered completely by C contamination and other impurities. Correspondingly to the identified chemical composition, the resulting colour of the sample was also dark, and does not represent the authenticity of the remaining material, i.e., the brass that was used to cover the wooden chest in the 4th century CE. EDX analysis did not detect the presence of Ti, Pb and Fe compared to the XRF and ICP-MS methods. The reason for this can be attributed to the fact that the EDX analysis is limited by the volume being analysed, and that it belongs to the semi-quantitative analysis methods with a fairly high degree of deviation which depend on the conditions of the analysis (acceleration voltage and current in the electron microscope, distance of the EDX detector from the sample, analysis site, magnification, etc.).

#### 3.2.2. SEM/EDX Microscopic Examination of the Polished Dark Sample

A macroscopic examination of the sample after removing the thin top layer showed the shine of the sample, while microscopically, it revealed a dendritic microstructure typical of the as-cast state of brass alloys—as shown in Figure 4. In the microstructure, dendrites with branch sizes of a few 100 µm are visible, distributed unevenly on the inspected polished surfaces. Due to the requirement to return the sample to the museum it was not possible to prepare in more detail metallographically, and thereby analyse the microstructure of the surface of the polished sample for a detailed examination using chemical etching and other approaches, since these do not preserve the authenticity of the original sample.

Brass is an alloy of copper and zinc where the Cu content is >65 wt.%. The properties of brass generally depend on the proportion of the two metals in the compound. The lower the proportion of zinc, the higher the melting point of the alloy, and the higher the ability to transform. The content of other alloying elements also affects the transformation properties of brass greatly. The characteristic of molten brass is that it has no tendency to gas and therefore pours well, and gas porosity does not occur in the microstructure. As the examination of the microstructure showed, no porosity and other casting defects were observed in it. We can conclude that the casting and transformation of the brass plate went without any significant problems in the 4th century CE. The chemical composition identified by EDX analysis on the polished surface is shown in Table 3. The content of Cu, Zn, Pb and Sn was determined after removing the top layer. The last two alloying elements listed are known from the literature as those added to brass due to requirements for higher hardness [10,11,12], and this was probably known to metallurgists at the time of the brass production. The examined microstructure of the brass plate was stable, without impurities and any inclusions, which is an important finding for the reconstruction process of the wooden chest. The EDX analysis revealed a significantly lower content of C and O compared to the content of the same elements on the unpolished brass plate. The fact is that the polishing process cannot remove the contamination and oxide layer completely; to achieve a perfectly clean surface we would have to use an Ar ion etching process, which was not allowed. Because of this, it was not possible to obtain a realistic chemical composition of the brass as the proportions of C and O were too high and affected the reduction in the proportions of the main components (Cu and Zn).

### 3.3. FTIR Spectroscopy

The recorded IR spectra of the adhesive were compared to the spectra in the available database. The spectral pattern matching results revealed that the analysed glue was most likely beeswax. Therefore, the IR spectra of commercially available beeswax were recorded and compared to the spectra of the unknown adhesives. Beeswax is a very complex organic material, consisting of over 300 different substances. The main components of the beeswax are usually esters, hydrocarbons and free fatty acids [13]. The chemical composition of the beeswax can vary significantly, depending on the bees’ species and degradation of the wax. Furthermore, the commercially available beeswax is often adulterated with the hydrocarbons of alien origin (e.g., paraffin) that alter the chemical composition [14]. The IR spectra of both samples of adhesive and the IR spectra of beeswax are presented in Figure 5a. As all the samples exhibited similar bands in the region around 3000 cm^−1^, corresponding to the CH_3_ and CH_2_ stretching vibrations, more careful examination of the spectra in the region from 2000 cm^−1^ to 600 cm^−1^ is presented in Figure 5b. The band at 1736 cm^−1^ was ascribed to C=O stretching vibrations of carboxylic groups in the ester linkage, the band at slightly lower wavenumbers (around 1710 cm^−1^) was ascribed to the C=O vibrations of the free carboxylic acids. The bands at 1473 cm^−1^ and 1376 cm^−1^ were ascribed to CH_2_ scissor deformation and CH_3_ symmetric deformation vibrations, respectively. In the region from 1290 cm^−1^ to 1040 cm^−1^ there were several absorption bands, with the band at 1171 cm^−1^ being the most prominent. This band was ascribed to C-O stretching vibrations of the esters. The band at 957 cm^−1^ was ascribed to C-H bending deformation vibrations. The band at 719 cm^−1^ was ascribed to CH_2_ rocking mode. The IR spectra of both adhesive samples were almost identical, while commercially available beeswax exhibited a slightly different spectrum.

### 3.4. DSC Analysis

It is well known from the literature [15] that beeswax exhibits two melting endothermic peaks upon heating: the first at around 56 °C and the second at around 68 °C. The thermal properties of samples determined by DSC are summarised In Table 4. In Figure 6a,b the DSC thermograms of the first cooling run and the second heating run are presented, respectively. All samples exhibited similar crystallisation and melting behaviour. Upon cooling, all samples exhibited double crystallisation exothermic peaks, the first (T_c1_) in the range of 54 °C to 55 °C and the second (T_c2_) in the range of 50 °C to 51 °C. Both adhesive samples exhibited the same exothermic peak shape, while the commercially available beeswax exhibited a slightly different peak shape. Upon heating, all samples exhibited double melting endothermic peaks: the first (T_m1_) in the range of 55 °C–56 °C and the second (T_m1_) in the range of 65 °C–66 °C. Moreover, both adhesive samples exhibited the same shape of endothermic peak for the cooling, while the commercially available beeswax exhibited a slightly different peak shape with the first melting peak being more prominent.

### 3.5. Solubility Tests

The adhesive samples were tested for solubility in three common organic solvents (acetone, n-hexane and dichloromethane), which were chosen based on the solubility of beeswax where data were found in the literature [16,17]. The purpose of the solubility tests was to determine the best solvent for delaminating the brass pieces from the glue on the existing plastic box without damaging and destroying the authenticity of the brass tiles.

Figure 7 shows vials containing different solvents and adhesive after 5 min of dissolution. The first number represents the solvent used and the second number represents the adhesive used. The pieces of adhesive dissolved in acetone were slightly swollen, but still present after the tested time. On the other hand, it was found that both n-hexane and dichloromethane dissolved the adhesives successfully, but with slight differences in solubility. Dichloromethane dissolved both adhesives fully, as indicated by the clear solution which only contained suspended impurities, while at the tested time, n-hexane dissolved the adhesives only partly, as indicated by the cloudy solution.

Based on the premise that it was possible to dissolve the adhesive that was used for gluing the brass tiles onto the plastic housing in 1973 with organic solvents, and thus remove the original brass tiles, the literature in the field of dissolving brass and related Cu compounds was reviewed in the considered organic solvents. A study of the available literature found that it was proven [18] that acetone remaining on the surface of Cu/Cu alloys after degreasing can react very slowly with water vapour under ambient light to form acetic acid and cause severe corrosion. Such a reaction is completely inhibited in the dark. This suggests that, when using acetone as a solvent medium for the glue, the experiment should be carried out in a dark room, with as little moisture as possible and in as short a time as possible, to avoid corrosion of the archaeologically important brass tiles.

Numerous studies of the corrosion behaviour of various metals and their alloys in natural seawater, fresh water and acidic environments are known. The scientific literature [19,20,21] discusses the physical and mechanical properties of brass in relation to the binary phase diagram of brass, the potential pH diagram of brass, the types of dezincification and the mechanism of dezincification. Investigations into the control of brass dezincification in various environments such as acidic, neutral and in the presence of contaminants, and the effect of adding alloying elements on reducing the dezincification process, are also discussed in more detail. There is nowhere to find research on the topic of dissolving brass in organic solvents such as n-hexane and dichloromethane. Even a short exposure test of a brass sample in both solvents, carried out in our own laboratory, did not show surface microstructure changes. Based on the above, we can conclude that the brass microstructure was practically stable to both types of organic solvents, and therefore we do not expect degradation or corrosion of archaeologically important brass tiles in real experimental dissolution.

In accordance with the completed study, a reconstruction proposal was made for the installation of a new chest (Figure 8). A suggestion related to the reconstruction is to make a test chest with samples of the same materials (brass and glue) before the reconstruction of the real chest itself. A variety of methods would be attempted to remove the brass plates from the adhesive on the existing Plexiglas chest, including:-By applying the solvent with a dropper or a brush;-By removing brass plates with a scalpel in combination with a solvent;-With immersion of individual brass plates in a solvent to finally remove the adhesive;-By thermal decomposition and mechanical solvent removal.

Based on the performed preliminary tests, a suitable final method will be determined for the removal of the archaeologically important brass tiles from the plastic chest in the Zadar Museum in Croatia (see Figure 8).

## 4. Conclusions

Based on the research carried out with the aim of reconstructing the existing chest in the Zadar Museum (Croatia) successfully, the conclusions are as follows:The investigation of archaeological objects is demanding, and therefore it is necessary to use non-destructive characterisation techniques to preserve the authenticity and archaeological value of the objects found;The results of our own investigations on samples of brass tiles and glue of a historically important chest confirmed the assumption that it is possible to remove brass tiles from an existing plastic chest with suitable solvents without any damage to the brass tiles;The proposal for the reconstruction of the chest includes the use of Dichloromethane with a dropper or a brush to the existing glue, so that the process results in 100% removal of all brass pieces. For those that cannot be removed, immersion in a bathtub is envisaged;The reconstruction and reproduction of the chest, which will reflect the authenticity of the 4th century CE, will require the cooperation, not only of material scientists, but also of archaeology and masters of superior manual skills. The removal of the old archaeologically historically significant brass plates from the plastic chest will require strict adherence to the removal protocol and great precision;For the preservation of cultural heritage, it is crucial to include different profiles of scientists in the research, in order to get as realistic an insight into the past as possible.

## Figures and Tables

**Figure 1 materials-16-00133-f001:**
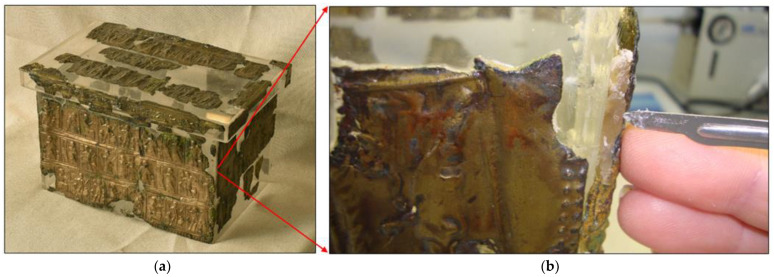
(**a**) Reconstruction of the wooden chest with brass tiles; (**b**) Presentation of taking samples from the chest (brass part and adhesive).

**Figure 2 materials-16-00133-f002:**
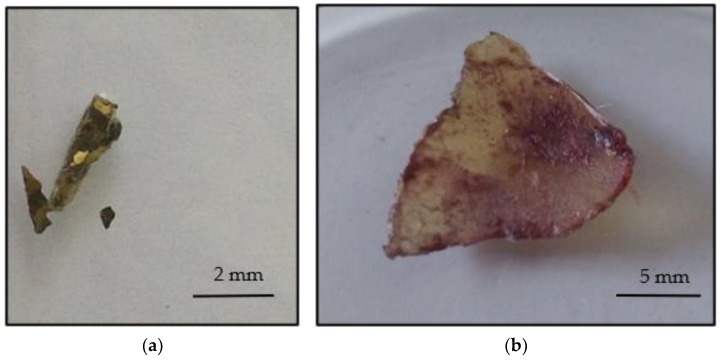
(**a**) Brass tiles and (**b**) Glue sample.

**Figure 3 materials-16-00133-f003:**
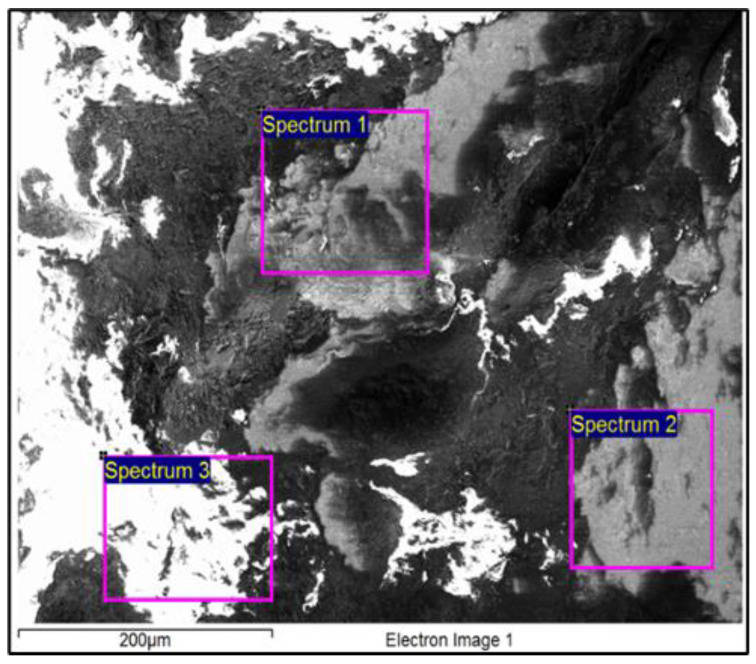
Microstructure of the unpolished dark sample’s surface with areas for EDX analysis.

**Figure 4 materials-16-00133-f004:**
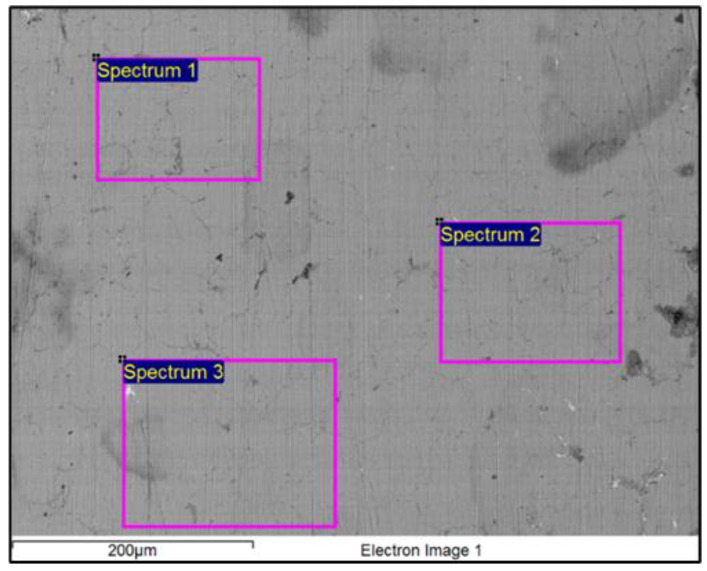
Microstructure of the polished dark sample’s surface with areas for EDX analysis.

**Figure 5 materials-16-00133-f005:**
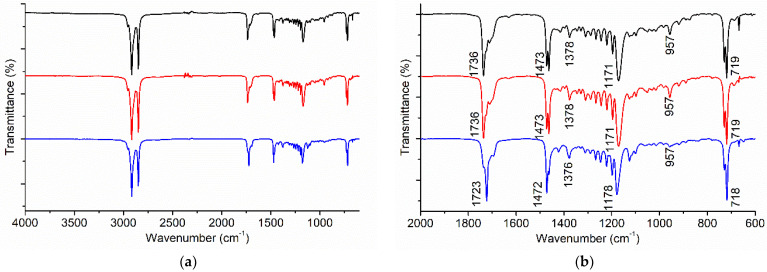
FTIR spectra of adhesive 1 (black), adhesive 2 (red), and commercial beeswax (blue): (**a**) Full spectra; (**b**) Region from 2000 cm^−1^ to 600 cm^−1^.

**Figure 6 materials-16-00133-f006:**
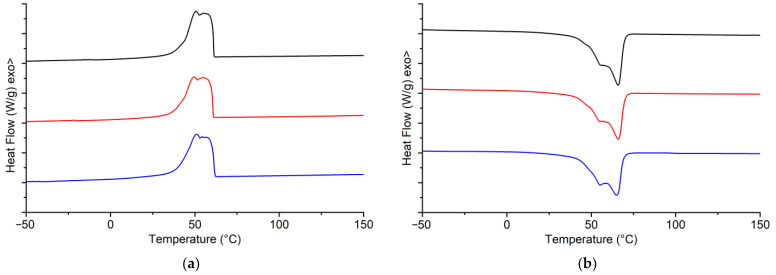
DSC thermograms of adhesive 1 (black), adhesive 2 (red), and commercial beeswax (blue): (**a**) Cooling run and (**b**) Heating run.

**Figure 7 materials-16-00133-f007:**
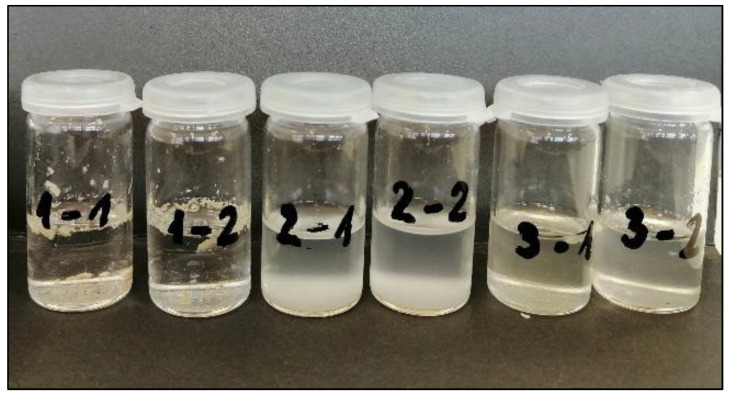
Solubility of the adhesives in acetone (1), n-hexane (2) and dichloromethane (3); the first number represents the used solvent, while the second number represents the used adhesive.

**Figure 8 materials-16-00133-f008:**
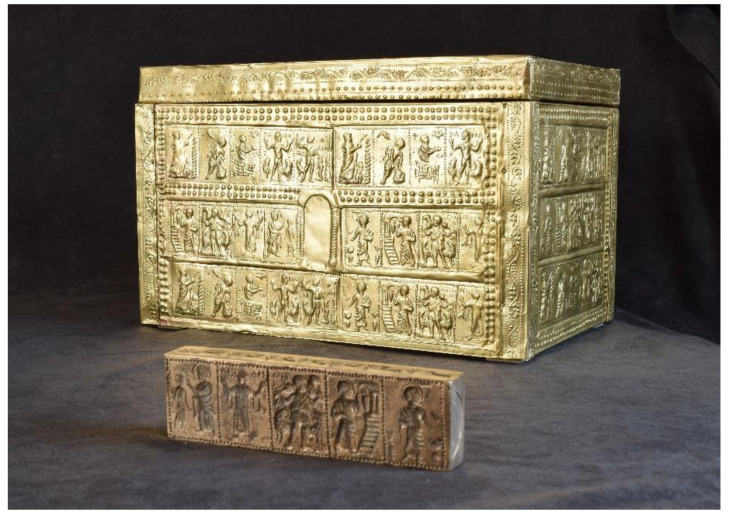
Presentation of a chest made of brass, where the sheet is glued to Plexiglas for the needs of the reconstruction of the existing chest in the Zadar Museum (Croatia).

**Table 1 materials-16-00133-t001:** Average chemical composition of the brass tiles measured by XRF and ICP-MS in wt.%.

**XRF**	**Sn**	**Zn**	**Fe**	**Ti**	**Pb**	**Cu**
Bright sample	0.99	6.6	1.2	0.32	12.9	rest
Dark sample	0.31	16.6	0.23	/	2.3	rest
Small sample	0.21	16.3	0.18	/	1.0	rest
**ICP-MS**	**Sn**	**Zn**	**Fe**	**Ti**	**Pb**	**Cu**
Small sample	0.13	8.7	0.11	<0.01	1.6	rest

**Table 2 materials-16-00133-t002:** Average chemical composition of the unpolished dark sample’s surface by EDX in wt.%.

EDX						
	C	O	Al	Si	Ca	Sn
mean	63.98	34.77	0.18	0.66	0.07	0.25
STD	2.41	2.15	0.27	0.39	0.16	0.23
Max	67.04	38.53	0.62	1.16	0.40	0.71
Min	59.55	31.81	0.00	0.00	0.00	0.00

**Table 3 materials-16-00133-t003:** Average chemical composition of the polished dark sample’s surface by EDX in wt.%.

	EDX					
	C	O	Cu	Zn	Pb	Sn
mean	5.50	2.10	61.45	23.34	7.05	0.56
STD	1.56	1.40	2.86	2.47	2.96	0.26
Max	8.85	4.51	64.61	26.24	11.13	1.24
Min	3.31	0.00	53.37	17.63	2.92	0.06

**Table 4 materials-16-00133-t004:** Thermal properties of adhesives sampled from two places of the box and commercial beeswax.

Sample	Adhesive 1	Adhesive 2	Beeswax
Mass (g)	13.3867	14.8614	14.2298
*T*_c1_ (°C)	55.4	54.7	54.2
*T*_c2_ (°C)	50.7	49.6	51.1
Δ*H*_c_ (J/g)	156.9	151.0	155.7
*T*_m1_ (°C)	56.0	55.3	55.1
*T*_m2_ (°C)	66.0	66.0	65.1
Δ*H*_m_ (J/g)	156.9	151.0	155.7

## Data Availability

Not applicable.

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
