# Peer review of "Material Analysis of the Remains of a Wooden Chest from the 4th Century and a Proposal for Its Reconstruction"

_materials, 2022, doi:10.3390/ma16010133_

Round 1
Reviewer 1 Report
Dear Authors,
The manuscipt submitted to review shows how materials science can be applied to the protection of cultural heritage. This is an extremely important issue that opens new perspectives for analysis and usage of materials for restoration and renovation of monuments.
Originally, reading the title, I thought about using composites imitating wood like unsaturated polyester resin filled with wood. But the work concerns the ancient brass plates built on a plastic chest - replacement of wood one.
The use of brass plates and glue combined with a "Plexiglass skeleton", allowed to obtain surprising results (Fig. 7). An important study was the selection of the appropriate glue solvent so as not to cause further damage.
Please expand Conclusions section and add more references (if they are).
I believe that the presented manuscript should be accepted for publication as presented because it brings something new that is hard to find.
Best regards
Author Response
Dear Reviewer,
Thank you for your review. The Conclusions section has been expanded, a few more references have been added.
We hope that the article is now suitable for publication.
Best Regards,
Yours faithfully,
Rebeka Rudolf
On behalf of the authors.

Reviewer 2 Report
This enjoyable read demonstrates the application of materials science in preserving culturally significant archaeological artefacts, in this case, brass plates circa the 4th century AD. The brass plates were mounted on a plastic box and fixed with an unknown adhesive. Through comprehensive characterisations, the authors could identify the brass alloy's composition and the adhesive used, which was beeswax. Then they proceeded by testing various solvents to deactivate the bond so that the brass plates could be safely removed and mounted on a more archive-worthy stand. They found acetone applied in dark and dry conditions can dissolve the adhesive while preserving the brass. Light, moisture, and prolonged exposure were predicted to corrode brass through dezincification. This work can set a protocol for similar preservation efforts, and I recommend it be published.
Author Response
Maribor, 2.12.2022
Dear Reviewer,
Thank you for a nicely written review.
Best Regards,
Yours faithfully,
Rebeka Rudolf
On behalf of the authors.

Reviewer 3 Report
See attached document.

Author Response
Maribor, 4.12.2022
Dear Reviewer,
Thank you for your review. The answers to your questions are highlighted in a red color.
Introduction, line 40-45: It is stated that the tiles belong to a ‘wooden chest made in the 4th century’. How do you know this? How was it dated? Did you find remaining wood, or how do you know that the tiles belong to a wooden chest? What do you mean with ‘expert examination’ – exist there a reference?
In accordance with the request, we have added a reference* in which the age of the chest in question is stated and explained.
*added reference:
ZBORNIK I. SKUPA HRVATSKE RANOKRŠÄ†ANSKE ARHEOLOGIJE (HRRANA) - PROCEEDINGS OF THE I. GROUP OF CROATIAN EARLY CHRISTIAN ARCHEOLOGY (HRRANA), 2020
A broader explanation as an answer to the question regarding the age of the chest is as follows:
The find of the chest panelling was the first to be scientifically processed by A. Badurina1. A brief description of the decoration on the panelling was provided by B. Ilakovac a little later2. Biblical images from the chest panelling were used to illustrate Croatin famous iconographic lexicon3. The depiction of the good shepherd from this chest proved to be important an element for interpreting the meaning of the depiction of shepherds in late antique art4. As an extremely significant monument, the chest is included in the manuals of ancient and early Christian archaeology5. It is dated between 330 and 350 AD6. According to Ivo Fadić's assumption, it was acquired in order to store a glass vessel and an undecorated silver reliquary in it, while a younger octagonal silver reliquary was added to the group a little later 7.
1 A. Badurina, Early Christian reliquary from Novalja, Archaeological problems on the Yugoslav coast of the Adriatic, IX. congress of archaeologists of Yugoslavia, Materials XII, Zadar 1976, 283–295, T. I–XVIII.
2 B. Ilakovac, Early Christian reliquaries of the Kesen (Cissa) diocese from Novalja on the island of Pag, Vjesnik Arheološkog muzeja u Zagrebu, 3.s., 26–27 (1993–1994), Zagreb 1994, 47–65.
3 A. Badurina, Lexicon of iconography, liturgy and symbolism of Western Christianity, 2. edition, Zagreb 1990.
4 N. Himmelmann, Über Hirten-Genre in der antiken Kunst, Opladen, Leverkusen 1980.
5 N. Cambi, Antika, 303, 304, fig. 479 Zagreb 2002.
6 Cambi, 51, note 179, 1994.
7 Fadić 1994, 164, 165.
Considering the topic of the Materials journal, we would not add it to the text of manuscript.
Line 79-93: this explanations are unnecessary and should be deleted.
Done
Line 104-105: unnecessary
Taken into account
Results: The results of elemental analysis should be presented more concisely. Where does the
titanium come from?
The results of the analysis of elemental analysis are as they are and no repetition is possible (sample is cultural heritage).
The titanium is probably a casting residue, because during the 4th century BC, casting technology was not yet developed enough to have a high-level purification process for the brass melt, which would ensure a high purity of the final alloy. We cannot confirm the direct source of Ti.
A review of other recent literature does not indicate that Ti is added to the brass alloy. More familiar cases are when it is added Ti to cast Cu-bronze alloy, because Ti is very effective in changing the microstructure of this alloys. In this way, a reinforcing effect of the bronze matrix is created, as well as surface modifications, which leads to improved surface properties of bronze alloys.
Line 256: SRF – a typo?
Corrected.
Line 257-261: Actually, the SEM-EDX analysis is more precise than the handheld XRF. A problem may be that ICP and SEM-EDX use only very small samples or areas, resp. which may not be representative for the entire object.
According to the requirements of the Ministry of Culture Croatia, Department for the Protection of Cultural Heritage, the proposed analysis for the entire object by the reviewer is not and will not be possible, because it is not allowed due to the extremely important cultural value of the found chest. Please read the introduction and the sample size that could be analysed with permission from all institutions.
Line 310: typo, blue instead of glue.
Corrected text.
Line 305-331: a figure with a comparison of pure beeswax would be helpful. However, IR spectra of waxes look all very similar, and to identify the single components and admixtures you would use GC-MS. I am in doubt of the detailed interpretation of the wax spectrum is necessary, probably a comparison with a reference spectrum would be more suitable.
The aim of the study in regard to characterisation of the adhesive was to identify its main chemical composition and related properties that are relevant for successful reconstruction, while the detailed identification of the individual components was out of the scope of the present study. Therefore, we decided to investigate the chemical composition of the adhesive using only Fourier transform infrared spectroscopy. The spectral pattern matching results revealed that the recorded spectra of the adhesives very closely resemble the spectrum of beeswax. To confirm this finding, we have purchased commercially available beeswax, recorded its spectrum, and compared it to the adhesives (comparison of pure beeswax and the adhesives is presented in Figure 5). Moreover, we have determined thermal properties of beeswax and adhesives using DSC analysis, that further increase our confidence in FT-IR interpretation, due to matching thermal properties. We agree that IR spectra of various waxes are relatively similar. However, the differences are still large enough to allow distinction of beeswax to other commonly used waxes, such as paraffin. In our opinion, the presented results and discussion are not very detailed interpretation of the wax spectrum, but rather the comparison to the reference spectrum (commercially available beeswax). However, we decided to ascribe the most characteristic absorption bands to the corresponding molecular vibrations, as in our opinion, this should provide value to the readers.
Solubility test: this chapter could be more concise. Just a question: if the adhesive is wax, could you warm up the object carefully, remove the tiles mechanically and clean them with hexane?
Response 2: The suggested method (warming up the tiles, mechanical removal, and solvent cleaning) would also be a viable approach for restoration.
We hope that the article is now suitable for publication.
Best Regards,
Yours faithfully,
Rebeka Rudolf
On behalf of the authors.

Reviewer 4 Report
I have reviewed the article, “Material analysis of the remains of a wooden chest from the 4th century and a proposal for its reconstruction” submitted by Rebeka Rudolf, Janez Slapnik and Rajko Bobovnik.
The paper reports an interesting analysis for the materials that used to glue valuable brass plates onto a box to serve as a reconstruction of a chest. It is now argued, in hindsight, that this reconstruction was poorly done, and should be “undone” and a new reconstruction undertaken.
The details of the analysis are well presented. However, important properties of the brass plates that justify a reconstruction are missing. Reference [1] is not useful. Have any of the owners of these plates subjected them to non-destructive trace-element analysis using Prompt Gamma Activation Analysis that could be performed at the reactor in Budapest.
As for publication, this manuscript appear to be better suited to a report to the leadership of the museum, rather than publication in a refereed journal….unless there are many such objects that will need to be deconstructed. If there are many such articles, something should be stated in the introduction.
Author Response
Maribor, 2.12.2022
Dear Reviewer,
Thank you for a review. Reference no. 1 cannot be removed because it represents the first professional mention of the chest, with key data at the level of the country of Croatia, where it was found. This reference was the basis for the protocol-based acquisition of all consents for conducting the research and the export of the "precious sample" from the country of the find to another country, Slovenia, where the investigation was carried out. Before that, practically nothing was written in the professional public "materials" about the issue of this chest.
Access to non-destructive analysis of trace elements using Prompt Gamma Activation Analysis, which would be carried out in Budapest, is practically impossible, since the Ministry of Culture of the Republic of Croatia does not have an agreement for the export of cultural heritage to the territory of Hungary.
References in the introduction have been added.
I hope that we have adequately answered and clarified matters.
Best Regards,
Yours faithfully,
Rebeka Rudolf
On behalf of the authors.

Round 2
Reviewer 3 Report
There is still a typo in line 241: 'Blue'
and in lines 242, 243: I suggest to use 'spectra' instead of 'spectrums'
Author Response
Maribor, 14.12.2022
Dear Reviewer,
Thank you for your review.
There is still a typo in line 241: 'Blue' → the text has been corrected
and in lines 242, 243: I suggest to use 'spectra' instead of 'spectrums'→ the text has been corrected
We hope that the article is now suitable for publication.
Best Regards,
Yours faithfully,
Rebeka Rudolf
On behalf of the authors.

Reviewer 4 Report
Reference 2 at least has English abstracts for the articles.
Author Response
Maribor, 14.12.2022
Dear Reviewer,
Thank you for your review.
Comment: Reference 2 at least has English abstracts for the articles.
Given the topic of the article, it is extremely difficult to provide all English references.
We hope that the article is now suitable for publication.
Best Regards,
Yours faithfully,
Rebeka Rudolf
On behalf of the authors.
